



# Tropical peat fire emissions: 2019 field measurements in Sumatra and Borneo and synthesis with previous studies

Robert J. Yokelson[1], Bambang H. Saharjo[2], Chelsea E. Stockwell[1,a], Erianto I. Putra[2], Thilina Jayarathne[3,b], Acep Akbar[4], Israr Albar[5], Donald R. Blake[6], Laura L. B. Graham[7], Agus Kurniawan[8,c], Simone Meinardi[6], Diah Ningrum[9], Ati D. Nurhayati[2], Asmadi Saad[10], Niken Sakuntaladewi[8], Eko Setianto[11], Isobel J. Simpson[6], Elizabeth A. Stone[3], Sigit Sutikno[9], Andri Thomas[7], Kevin C. Ryan[12], Mark A. Cochrane[13]

[1]University of Montana, Department of Chemistry, Missoula, 59812, USA

[2]IPB University, Faculty of Forestry and Environment, Bogor, 16680, ID

[3]University of Iowa, Department of Chemistry, Iowa City, 52242, USA

[4]Ministry of Environment and Forestry, Banjarbaru Environment and Forestry Research Institute, Banjarbaru,70721, ID

[5]Ministry of Environment and Forestry, Directorate General of Climate Change, Jakarta 10270, ID

[6]University of California, Irvine, Department of Chemistry, Irvine, 92697, USA

[7]Borneo Orangutan Survival Foundation, Mawas Program, Bogor 16128, ID

[8]Ministry of Environment and Forestry, ID

[9]Riau University, Center for Disaster Research, Pekanbaru, ID

[10]Jambi University, Faculty of Agriculture, Jambi, ID

[11]KKI-WARSI, Jambi, 36361, ID

[12]FireTree Wildland Fire Sciences, L.L.C., Missoula, 59801, USA

[13]University of Maryland Center for Environmental Science, Appalachian Laboratory, Frostburg, 21532, USA

[a]now at: University of Colorado, Cooperative Institute for Research in Environmental Sciences, Boulder, 80309, USA and NOAA, Chemical Sciences Laboratory, Boulder, 80305, USA

[b]now at: Bristol Myers Squibb, New Brunswick, 08902, USA

[c]now at: National Research and Innovation Agency, Center for Research and Development of Biotechnology and Forest Plant Breeding, Yogyakarta, 55582, ID

*Correspondence to*: R. J. Yokelson (bob.yokelson@umontana.edu)





**Abstract**

Peat fires in Southeast Asia are a major source of trace gases and particles to the regional-global atmosphere that influence atmospheric chemistry, climate, and air quality. During the 2015 November record-high Ocean Niño Index (ONI, 2.6) our mobile smoke sampling team made the first, or rare, field measurements of numerous trace gases,

aerosol optical properties, and aerosol chemistry and mass emissions for fires burning only peat in the Indonesian province of Central Kalimantan (on the island of Borneo). The measurements used Fourier transform infrared spectroscopy (FTIR), whole air sampling (WAS), photoacoustic extinctiometers (PAX, 401 and 870 nm), and detailed off-line analyses of particulate matter (PM) collected on filters. In September-November 2019 we measured peat fire trace gas emissions again, using WAS only, under ENSO-neutral conditions (ONI, 0.3) in more remote areas of Central

Kalimantan and also the Indonesian provinces of Riau, Jambi, and South Sumatra, all on the island of Sumatra. The 2019 measurements significantly expanded the geographic range and climate conditions sampled. This paper presents the 2019 results and synthesizes them with the previous field work to converge on more robust regional average emission factors (EFs, g compound per kg biomass burned) for authentic peat fires. In addition, samples of peat imported from Indonesia were burned in U.S. laboratories and the EFs and optical properties were characterized in

more detail than in the field by a larger suite of instrumentation. We use the improved knowledge of regional emissions based on the expanded field measurements to select the most representative lab data and compute a synthesized, more "chemically-complete" set of EFs and aerosol optical properties for tropical peat fires.

The modified combustion efficiency (MCE) values for the peat smoke sampled in 2019 were within the range of MCEs sampled in 2015, but with a lower average in 2019 ($0.718 \pm 0.021$, range $0.687 - 0.736$) than 2015 ($0.772 \pm$

$0.035$, range $0.693 - 0.835$). Averaging the new and older data together suggests an updated MCE for tropical peat fires of ~0.76. Despite the difference in MCE, the study-average methane emission factors (EF $CH_4$) were remarkably similar across the two years probing different regions: $9.42 \pm 2.51$ g $kg^{-1}$ in 2019 and $9.51 \pm 4.74$ g $kg^{-1}$ in 2015. When parsing the 2019 samples by province, the EFs for non-methane organic gases (NMOG) were about 3 times higher in South Sumatra and Central Kalimantan than in Jambi and Riau, but the overall 2019 study average was only ~15%

higher than the 2015 study average. South Sumatra peat fires emitted higher amounts of carbonyl and dimethyl sulfide, suggesting a volcanic or marine influence or effects of agricultural chemicals. The lab and field work taken together provide EFs for 230 trace gases including $CO_2$ ($1538$ g $kg^{-1}$), CO ($305$ g $kg^{-1}$), and $CH_4$ ($10.3$ g $kg^{-1}$). These are significant adjustments to IPCC-recommended EFs, -10%, +45%, and -49%, respectively. We also report EFs for numerous NMOG, 46 N-containing compounds, and 14 sulfur or halogen-containing species. The use of high-

resolution mass spectrometry in the lab allowed measurement of 82% more NMOG mass than in the field. Gravimetrically measured EF $PM_{2.5}$ in the field in 2015 ($17.3 \pm 5.8$ g $kg^{-1}$) was ~20% lower than the average from lab studies ($22.4 \pm 10.4$ g $kg^{-1}$) perhaps due to higher field temperatures. Taken together the lab and field data show that the single scattering albedo (SSA) was largely independent of wavelength and MCE in the visible (~0.998), but lower at low MCE at 401 and 405 nm with a value of 0.958 at the study-average MCE. The absorption Ångström exponent

(AAE) at the average MCE was 5.7. By far the largest PM component was weakly-absorbing insoluble organic carbon.



## 1 Introduction

Global peatlands store an estimated 500-700 gigatonnes of carbon (GtC), which is similar in mass to the global atmospheric carbon pool (~850 GtC) and ~20-30% of the global terrestrial carbon mass (Warren et al., 2017; Watson et al., 2019; Turetsky et al., 2015). About 15% of global peat is located in the tropics, and about 41% of tropical peatland and 65% of tropical peat carbon is located in southeast Asia where peatland changes due to climate-change induced rainfall reduction, draining, and subsequent fires are currently the greatest (Warren et al., 2017; Dargie et al., 2017; Fatoyinbo et al., 2017). Deshmukh et al. (2021) found that draining Sumatran peatland decreased $CH_4$ emissions from anaerobic decomposition, but increased $CO_2$ emissions from aerobic decomposition and fluvial export of carbon. Draining also increased $N_2O$ emissions, potentially by accelerated mineralization of the peat under aerobic conditions producing $N_2O$ as a by-product. The net effect of draining was increased global warming. Climate-change induced reductions in precipitation could also increase peatland greenhouse gas (GHG) emissions by similar mechanisms. Reduction in the water table by draining or reduced rainfall also promotes fire, which converts semi-fossilized peat fuel and other biomass into $CO_2$, $CH_4$, and many other trace gas and aerosol species (Stockwell et al., 2016a; Vetrita et al., 2021; Sinclair et al., 2020). Fire plus non-fire GHG emissions associated with draining peatlands are greater per unit area than for any other land use change considered by the IPCC (Warren et al., 2017).

Peatland fires cause a broad suite of other impacts as well. The direct effect of aerosol emissions can offset GHG warming depending on their optical properties (Stockwell et al., 2016a; Lee et al., 2018a; Eck et al., 2019; Pokhrel et al., 2016; Liu et al., 2014) and the aerosols also impact cloud cover (Ding et al., 2021) and rainfall (Hodzic and Duvel, 2018; Chen et al., 2017; Lu and Sokolik, 2013). The aerosols and gases emitted by southeast Asian peatland fires are extensive enough to impact air quality regionally (Aouizerats et al., 2015; Hansen et al., 2019; Kiely et al., 2020; Koplitz et al., 2016; Lee et al., 2018b; Tosca et al., 2011; etc.). On a larger scale, volatile organic compounds (VOCs) from the 2015 Indonesian peatland fires had wide-ranging, significant impacts on the chemistry of the upper troposphere and lower stratosphere (Rosanka et al., 2021) and southeast Asian fires can contribute to trans-Pacific ozone transport (Xue et al., 2021).

Despite peat fires in the Indonesian provinces on the islands of Sumatra, Kalimantan, and Papua, and in Malaysian Borneo being a major, global atmospheric source of trace gases and particles (Akagi et al., 2011; van der Werf et al., 2010), until recently our knowledge of the emissions was limited to the results from burning one sample of peat from South Sumatra in a laboratory study (Christian et al., 2003). In 2012 three peat samples from Kalimantan were burned, also in a laboratory study, and the emissions were sampled with an extensive suite of state-of-the-art instrumentation (e.g., Stockwell et al., 2014; 2015; Jayarathne et al., 2014; Hatch et al., 2015; 2017). Some significant differences were observed in the emissions between the two lab studies and the lack of detailed field measurements at that time made it difficult to ascertain any potential regional differences or determine the most representative tropical peat fire data (Stockwell et al., 2014).

In October-November of 2015, as part of an extensive peat fire study that included investigations of land use and fire history, fuels mapping, remote sensing, lidar terrain transects, and a large hydrology component; we conducted ground-based field measurements of trace gases and aerosols in numerous peat fire plumes near Palangka Raya, Central Kalimantan (Applegate et al., 2012; Ichsan et al., 2013, Graham et al., 2014a, b; Hooijer et al., 2014; Stockwell



et al., 2016a; Jayarathne et al., 2018). We measured trace gas emission factors (EFs, g compound produced per kg peat burned) for ~90 gases using a Fourier transform infrared spectrometer (FTIR) and whole air sampling (WAS) canisters analyzed by gas chromatography (GC). Using photoacoustic extinctiometers (PAX) we measured EFs for scattering and absorption coefficients (EF $B_{scat}$, EF $B_{abs}$, $m^2$ $kg^{-1}$ peat burned) at 870 and 401 nm, the single scattering

albedo (SSA) at 870 and 401 nm, the absorption Ångström exponent (AAE), and EFs for black carbon (BC), etc. (Stockwell et al., 2016a). The filter samples provided EFs for elemental carbon (EC), organic carbon (OC), $PM_{2.5}$, metals, water-soluble ions, and numerous organic aerosol constituents such as PAHs and tracers (Jayarathne et al., 2018). This work provided the first reasonably complete field measurements of the emissions from burning the peat component of authentic peatland fires and provided important updates for peat fire EFs, but it was limited to samples

from one province under extreme drought conditions as revealed by the all-time record high value of the Ocean Niño Index (ONI) during the sampling (2.6, https://origin.cpc.ncep.noaa.gov/products/analysis_monitoring/ensostuff/ONI_v5.php, last accessed 26/4/2021).

Following our 2015 field study, another large-scale comprehensive lab experiment included Kalimantan peat fuel in 2016 (e.g., Selimovic et al., 2018; Koss et al., 2018), separate lab peat fire results were reported by Watson et al.

(2019), and Smith et al. (2018) reported field measurements of a suite of trace gases emitted by Malaysian peat fires in 2015-2016.

To address the limited geographic range of tropical peat fire field measurements, in September – November of 2019 we outfitted sampling teams with our most mobile sampling technique (WAS) to facilitate sampling across three provinces of Sumatra and a more remote area of Central Kalimantan than was sampled in 2015. Twenty-five fires

burning just peat (i.e. no surface vegetation contribution) were successfully sampled under ONI neutral conditions (0.3) at sites reflecting a large variety of land uses. In this paper we report the 2019 field results and compare them to the previous field results. We derive a more robust regional average set of tropical peat fire EFs based on our 2015 and 2019 field studies, literature EF for other field-sampled peat fires in peninsular Malaysia and Kalimantan, and a carefully selected subset of laboratory peat fires. We close by providing updated context and guidance for

implementing EFs in atmospheric models.

## 2 Experimental details

### 2.1 Site descriptions

Peat is partially decayed organic matter that, in the tropics, historically most often accumulated in evergreen peat swamp forests (Page et al., 2002). Undisturbed it can be classified as fibric, hemic, or sapric as depth, degree of

decomposition, and density all increase (Wüst et al., 2003). However most tropical peat fires now occur on sites disturbed by various types of agriculture, logging, dredging for canals, road building, and previous fires and also abandoned post-agriculture sites making traditional classification schemes less applicable. Given this complex environment, we targeted sampling peat fires in as wide a variety of locations as possible. We sampled 25 fires over a three-month period on sites with a variety of land-use trajectories ranging from working rubber plantations to

abandoned land dominated by shrubs, ferns, or second-growth forest. The map in Fig. 1 showcases the wide





geographic distribution of the sampling sites in regional context. The province, site name, date, number of samples, land use notes, and an emissions metric are shown in Tab. 1. More extensive site details including peat depth, geo-location, weather, etc. are found in Tab. S1. Detailed maps, photos, and additional data and calculations are in the open-access project archive (https://tinyurl.com/yc6yhvx7).

**Table 1.** Summary of sites and plumes sampled in 2019. The ΔNMHC / ΔCO ratio is shown for each plume and also (in bold) based on a plot including all the samples in a province.

| Province | Date | n | Land use notes | NMHC/CO | $R^2$ |
|---|---|---|---|---|---|
| Site name | dd/mm/yyyy | | | ppt/ppb | |
| **Jambi** | | **22** | | **12.2** | **0.731** |
| Desa Puding | 2/10/2019 | 7 | mix palm oil and brush | 15.77 | 0.741 |
| PT BEP | 2/10/2019 | 3 | mix palm oil and brush | 9.32 | 0.503 |
| Tahura | 3/10/2019 | 8 | ferns (burned in 2015) | 20.15 | 0.949 |
| PT ATGA | 6/9/2019 | 4 | palm oil (burned in 2015 | 12.65 | 0.706 |
| **Riau** | | **12** | | **14.26** | **0.987** |
| Desa Rimbo Panjang Kampar | 4/9/2019 | 2 | recent palm oil | 5.45 | 1 |
| Desa Rimbo Panjang Kampar | 30/9/2019 | 3 | recent palm oil | 12.68 | 0.9997 |
| Desa Manunggal Kampar | 4/9/2019 | 4 | palm oil | 14.13 | 0.997 |
| Desa Bukit Timah Dumai | 1/10/2019 | 3 | abandoned, grass and brush | 17.67 | 0.9995 |
| **South Sumatra** | | **24** | | **35.23** | **0.835** |
| Tempirai | 8/10/2019 | 2 | abandoned land, shrubs, ferns, small trees | 13.3 | 1 |
| Kayulabu | 8/10/2019 | 2 | abandoned land, shrubs, ferns, small trees | 31.92 | 0.997 |
| Kayulabu | 9/10/2019 | 2 | abandoned land, shrubs, ferns, small trees | | |
| Senasi Mulya | 10/10/2019 | 4 | abandoned land, shrubs, ferns, small trees | 15.28 | 0.999 |
| Tempirai | 11/10/2019 | 5 | abandoned land, shrubs, ferns, small trees | 22.17 | 0.856 |
| Senasi Mulya | 9/11/2019 | 2 | abandoned land, shrub, grass, small trees | 26.71 | 1 |
| Senasi Mulya | 10/11/2019 | 3 | abandoned land, shrub, grass, small trees | 61.71 | 0.982 |
| Senasi Mulya | 12/11/2019 | 4 | abandoned land, shrub, grass, small trees | 23.42 | 0.803 |
| **Central Kalimantan** | | **23** | | **38.25** | **0.692** |
| Canal wetland | 12/10/2019 | 2 | abandoned land, ferns, shrubs, trees | 7.62 | 1 |
| Canal Bapak Rista | 13/10/2019 | 4 | mix above and rubber plantation | 31.27 | 0.973 |
| Canal Bapak Rista | 14/10/2019 | 1 | mix above and rubber plantation | | |
| Canal Jayanti | 14/10/2019 | 3 | abandoned land, ferns, shrubs, trees | 42.49 | 0.999 |
| Canal Jayanti | 15/10/2019 | 3 | mix above and rubber plantation | 52.76 | 0.678 |
| Garitik | 29/10/2019 | 2 | abandoned land, some small trees | 8.4 | 1 |
| Garitik | 30/10/2019 | 6 | abandoned land, some small trees | 15.73 | 0.964 |
| Garitik | 2/11/2019 | 2 | abandoned land, some small trees | 15.94 | 0.826 |



**Figure 1.** Location of 2019 peat fire sampling sites.





### 2.2 Instrument descriptions and calculations

All the instrumentation, sampling strategies, and calculations pertinent to this study have been described in full previously (Stockwell et al., 2016a; Jayarathne et al., 2018). Here we briefly summarize the WAS approach and EF calculations. We note that after the 2015 field study was published (Stockwell et al., 2016a), the nominal 405 nm

wavelength in one PAX was measured more precisely to be 401 nm, which is updated in this work.

### 2.2.1 Whole air sampling (WAS) in canisters

Previously-evacuated 2 L stainless steel canisters were opened and filled quickly to ambient pressure directly in peat fire smoke plumes or adjacent background air. The canisters were then closed and shipped to the University of California, Irvine for measurement of a large number of gases (Simpson et al., 2006). Species quantified included

$CO_2$, CO, $CH_4$, and up to 100 non-methane organic gases (NMOGs) by gas chromatography (GC) coupled with flame ionization detection, electron capture detection, and quadrupole mass spectrometer detection as discussed in detail by Simpson et al. (2011). About ~70 of the NMOGs are combustion products that were enhanced in the source plumes and reported here. $CO_2$, CO, and $CH_4$ data have an uncertainty of a few percent. The limit of detection for most NMOGs was ~10 pptv or better, usually several hundred times below the concentrations that were sampled. The

precision and accuracy vary by compound or compound class and are reported in Simpson et al. (2011). Styrene is known to decay in canisters and the styrene data should be taken as lower limits. Our sampling strategy purposely targeted sampling many fires to characterize variability rather than intensive characterization of fewer fires. One background sample upwind of the fire and 1-3 smoke samples in the plume was typical.

### 2.2.2 Emission ratio and emission factor determination

The samples from each province were treated as a group. Within each of the four groups, the molar emission ratio (ER, e.g. $\Delta X/\Delta CO$) for all the WAS species X relative to CO was calculated by linear regression. EFs were computed from the complete set of ERs, by the carbon mass balance method, which assumes all major carbon-containing emissions have been measured (Ward and Radke, 1993; Yokelson et al., 1996, 1999):

$$EF(X)\left(g\,kg^{-1}\right) = F_C \times 1000 \times \frac{MM_x}{AM_C} \times \frac{\dfrac{\Delta X}{\Delta CO}}{\sum_{j=1}^{n}\left(NC_j \times \dfrac{\Delta C_j}{\Delta CO}\right)} \tag{1}$$

where $F_C$ is the carbon mass fraction of the fuel; $MM_x$ is the molar mass of species X; $AM_C$ is the atomic mass of carbon (12.01 g mol$^{-1}$); $NC_j$ is the number of carbon atoms in species j; and $\Delta C_j$ or $\Delta X$ referenced to $\Delta CO$ are the molar ERs for the respective species. We assumed a carbon fraction ($0.579 \pm 0.025$) measured earlier as the average of seven samples of Kalimantan peat (ALS Analytics, Tucson) (Stockwell et al., 2014). EFs are proportional to assumed carbon content, making future adjustments to EFs trivial if warranted based on additional carbon content

measurements. The denominator of the last term in Eqn. (1) estimates total carbon emissions, which we derived from summing the carbon in all the gases measured by WAS. Ignoring the carbon emissions not measurable by WAS (OC,



BC, unmeasured NMOG) likely inflates the EF estimates by less than ~5 % (Yokelson et al., 2013; Stockwell et al., 2015).

The relative amount of smoldering and flaming combustion during a biomass fire is often estimated from the modified combustion efficiency (MCE). MCE is defined as the ratio $\Delta CO_2/(\Delta CO_2+\Delta CO)$ and is mathematically equivalent to

$1/(1+\Delta CO/\Delta CO_2)$ (Yokelson et al., 1996). In the case of peat fires, all the combustion is by what is often simply termed as smoldering combustion. However, in the analysis of these fires, it is worth considering that "smoldering" actually refers to a mix of distillation of volatiles, pyrolysis of biomass (producing mainly a large variety of NMOGs and organic aerosol), and gasification of char (producing mainly $CH_4$, $NH_3$, CO, $CO_2$, $H_2$, and little visible aerosol) (Yokelson et al., 1996; 1997). Nonetheless, MCE can still be used to explore variability and MCE may vary with the

ratio of glowing combustion to pyrolysis (Yokelson et al., 1997).

### 3 Results and Discussion

#### 3.1 Trace gas emission factors measured in the field

In the 2015 field work the FTIR provided the capability for essentially unlimited real-time or grab sampling. Between the FTIR and WAS a total of 333 grab samples were spread over 35 distinct plumes and we calculated ER and EF for

each plume/fire. In 2019 a total of 81 WAS canisters were used to sample fires in four provinces limiting us to fewer samples per fire and leading us to explore consolidating the data by other factors. Each province was sampled by a dedicated team and we found that grouping samples by province produced highly correlated ER plots with distinct province to province differences. To explore the impact of the analysis approach on study conclusions we used the ratio of total measured non-methane hydrocarbons (NMHC) to CO since the NMHC were the most variable major

emission (vide infra). Table 1 shows the $\Delta NMHC / \Delta CO$ ratios from plots based on all the samples in a province and from plots based on just the samples in each plume. Provincial averages based on all the individual plumes were not statistically different from averages based on consolidated data. The study average and variability for the four provincial averages based on consolidated data ($25 \pm 14$, $1\sigma$) was similar to the study average and variability computed based on all the individual plumes ($22 \pm 15$). No clear patterns emerged when consolidating samples by vegetation

type or land-use. For instance, some fern-covered abandoned land tended to have high $\Delta NMHC / \Delta CO$ ratios, but other nominally similar sites did not. Since our study focus was spatial and interannual variability, we opted to report EFs for each of the four provinces for the 2019 samples, but all our raw mixing ratios and explicit EF calculations are available in our open access archive (https://tinyurl.com/yc6yhvx7) should others wish to pursue additional analyses. Note that provincial averages based on more sampling or a detailed knowledge of the spatial and temporal distribution

of fire uses/characteristics could be different in this highly complex environment. Table S2 presents the full set of MCEs and EFs calculated for 2019 alongside the Kalimantan field data from 2015 for the same species. Next, we describe the main features of the MCEs and EFs from the four provinces sampled in 2019 and compare them to the MCEs and EFs from 2015.

$CH_4$ is the second most important greenhouse gas emitted by peat fires after $CO_2$. We plot EF $CH_4$ versus MCE in

Fig. 2 to provide a good overview of both regional peat fire $CH_4$ emissions and the annual and interannual variability





in MCE. In Fig. 2, the black symbols show the context from our 2015 field work. MCE ranges from 0.693 to 0.835 and EF $CH_4$ ranges from 3.7 to 22.8 g kg$^{-1}$, averaging 9.51 ± 4.74 g kg$^{-1}$ (Stockwell et al., 2016a). Shown in green are additional EF $CH_4$ from previous sampling of 10 peat fire plumes in Malaysia in 2015-2016 by Smith et al. (2018). The Smith et al. samples have MCEs that overlap the upper two-thirds of the Stockwell et al. (2016a) samples. The

Smith et al. EF $CH_4$ are within the Stockwell et al. (2016a) range except for one high value of 26 g kg$^{-1}$, and they have a study average of 11 ± 6 g kg$^{-1}$, similar to the Stockwell et al. (2016a) study average. Hamada et al. (2013) and Huijnen et al. (2016) each report a study-average MCE and EF $CH_4$ based on limited sampling of peat fires in Central Kalimantan in 2009 and 2015, respectively. These values also lie in the range reported by Stockwell et al. (2016a). Against this backdrop, our 2019 "provincial average EF" are shown in red. Our 2019 MCEs overlap the lower one-

third of the Stockwell et al. (2016a) samples ranging from 0.687 to 0.736. The 2019 EF $CH_4$ are relatively tightly clustered around the average of 9.42 ± 2.51 g kg$^{-1}$. Thus, a picture emerges of highly variable $CH_4$ emissions, but with a robust, reproducible average based on field data of 10.3 ± 2.5 g kg$^{-1}$.

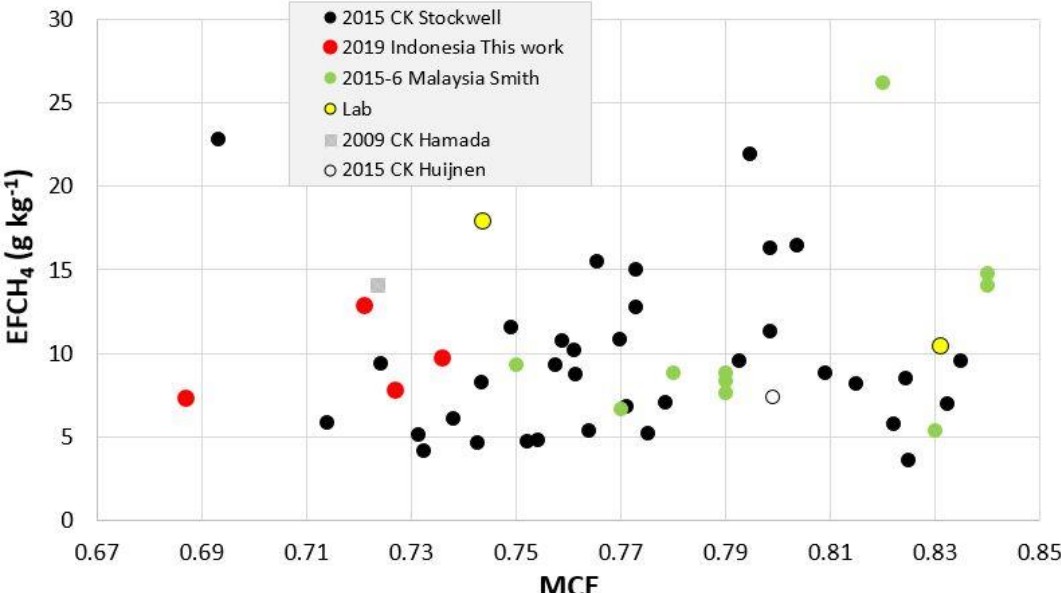

**Figure 2.** The emission factor for $CH_4$ (g kg$^{-1}$) versus MCE for field studies sampling pure peat fires in the tropics and lab studies

of pure tropical peat smoke that also included extensive NMOG data. "CK" indicates the Indonesian province Central Kalimantan. Black Stockwell et al. (2016a), red this work (provincial averages), green Smith et al. (2018), gray Hamada et al. (2013), unfilled circle Huijenen et al. (2015). Lab data (yellow) shown are from Fire #55 in FIREX (MCE = 0.831) and Fire #114 in FLAME-4 (MCE = 0.744). See text for additional details.

Next, we turn our attention to an overview of the NMOG emissions. Fig. 3 shows a plot of the 2019 NMOG EFs from

Central Kalimantan, South Sumatra, and Riau versus the 2019 NMOG EFs from Jambi, which had the lowest EFs. The division of the provinces into a high NMOG EF and low NMOG EF group is apparent. The Riau NMOG EFs were only slightly higher than Jambi (slope 1.27, r$^2$ 0.932). In contrast, both South Sumatra (slope 2.70, r$^2$ 0.936) and Central Kalimantan (slope 2.77, r$^2$ 0.935) had NMOG EFs almost 3 times larger on average. The slopes are similar,



but with lower $r^2$ when restricting the analysis to EFs < 0.3 g kg$^{-1}$. Interestingly, the high and low provinces combine to generate 2019 study-average NMOG EFs that are only about 15% higher than the 2015 study-average NMOG EFs in Stockwell et al. (2016a) (slope 1.15, $r^2$ 0.829) across 57 co-measured species as shown in Fig. 4. Restricting the analysis to EFs < 0.35 g kg$^{-1}$ lowers the $r^2$ to 0.7 and increases the slope to 1.6. Note that we have included highly

5    variable sulfur compounds (see Sect. 3.2) in these plots. Overall, a picture emerges of highly variable emissions, but fairly stable regional averages as additional data become available. Finally, in Fig. 5, we compare the 2019 NMOG EFs from Central Kalimantan to the 2015 NMOG EFs from Central Kalimantan (slope 1.63, $r^2$ 0.781). This gives some measure of the variability to be expected within the same province, but across different years with different drought conditions (ONI 2.6 in 2015 and 0.3 in 2019) and at different levels of disturbance since the 2019 samples

10    were in a less-disturbed, more remote section of the province accessible only by boat.

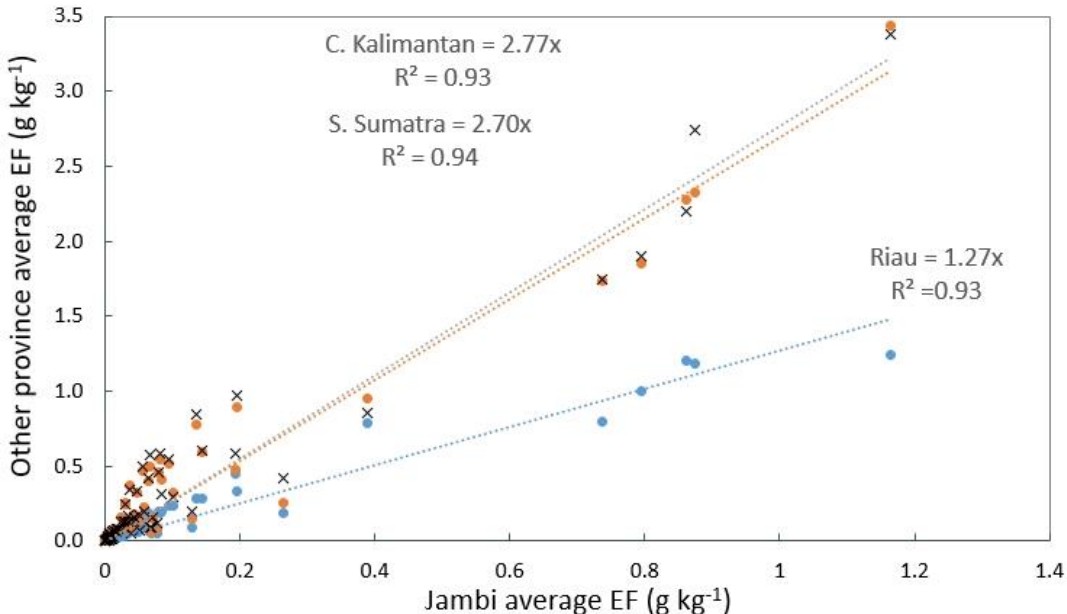

**Figure 3.** NMOG EFs calculated from WAS measurements in 2019 plotted for three provinces versus Jambi province, which had the lowest EFs on average. Riau (blue) EFs are about 27% larger than Jambi in this framework and Central Kalimantan (grey) and South Sumatra (orange) EFs are about 2.7 times larger.





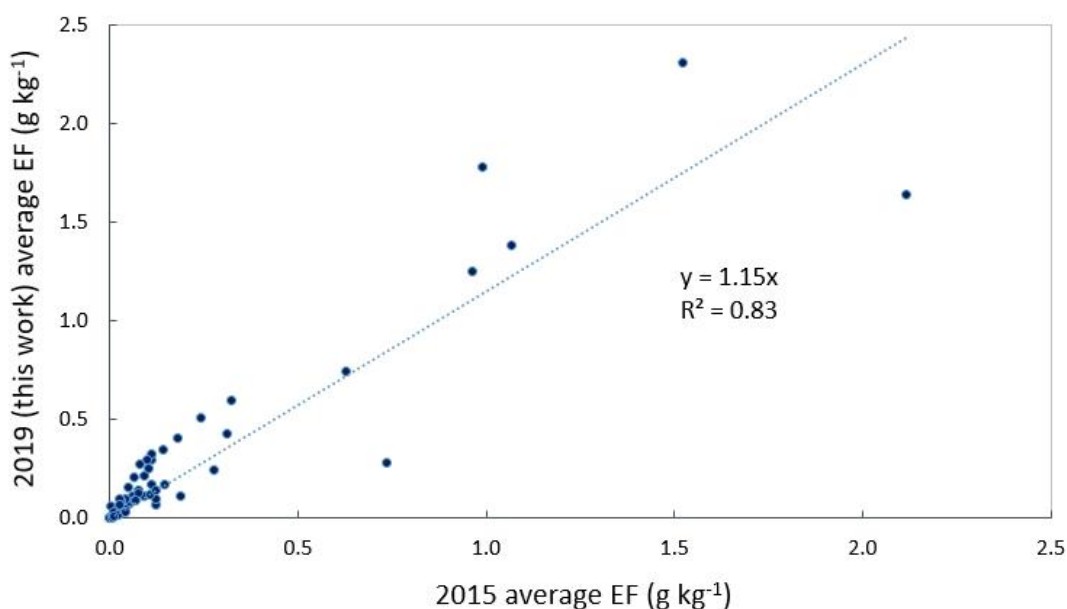

**Figure 4. The** 2019 study-average NMOG EF (this work, 4 provinces including Central Kalimantan) versus the 2015 study-average NMOG EF measured by Stockwell et al. (2016a) in Central Kalimantan for the 57 species measured in both studies.

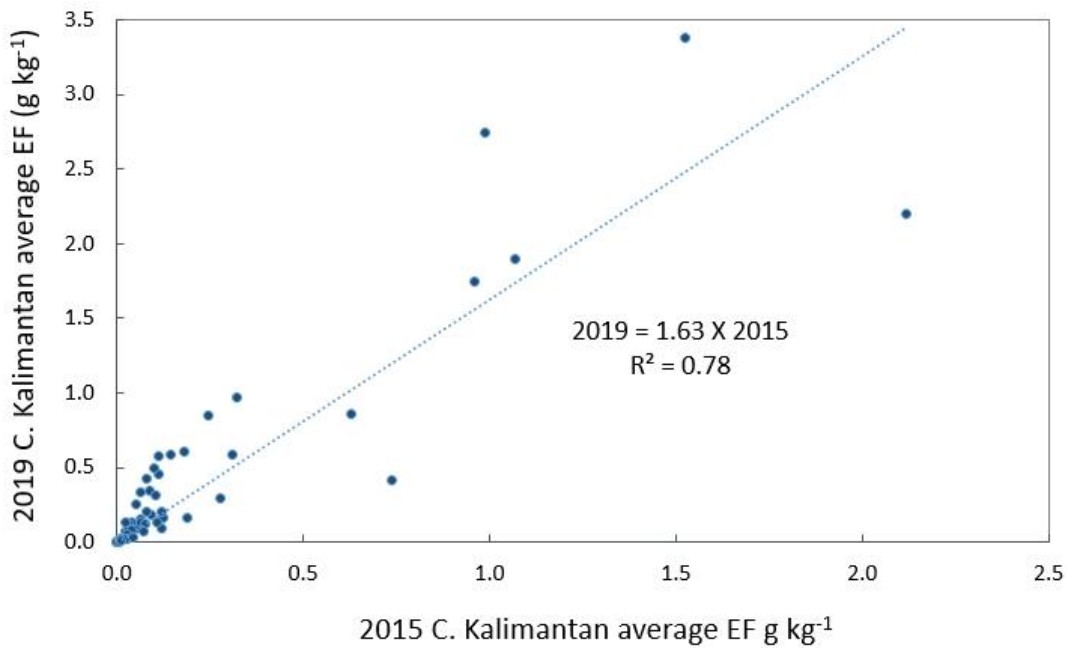

**Figure 5.** NMOG EFs (g kg$^{-1}$) measured in 2019 in Central Kalimantan (this work) versus NMOG EFs measured in Central Kalimantan in 2015 by Stockwell et al. (2016a).



### 3.2 Selection of representative lab data and discussion of trace gas EFs

Stockwell et al. (2016a) and others compared and synthesized data from previous lab and field studies and noted the large amount of important NMOG data added by on-line mass spectrometry, which is so far only available in lab studies. With the enhanced knowledge of the range of emissions from real-world peat fires provided by 2009-2019

field measurements we can re-assess which laboratory peat fire data are most representative of authentic peat fires. Table S3 shows the EFs from the previously-discussed field work along with selected lab studies. In particular, the oldest study by Christian et al. (2003) stands out as being on the extreme upper edge of field-observed MCE, consistently having a factor of two or more higher EFs for many compounds, and relying on a single sample. An examination of old photos also reveals that the peat sample burned was more powdery than the samples in subsequent

studies, perhaps due to shipping damage.

Eleven years after the Christian et al. study, nine peat fires were burned during the Fourth Fire Lab at Missoula Experiment (FLAME-4) in 2012 with more comprehensive emissions measurements provided by a large-scale multi-investigator team (Stockwell et al., 2014; 2015). Three fires each were burned in Canadian peat (#s 69, 112, 124), North Carolina peat (#s 61, 113, 150), and Indonesian (Kalimantan) peat (#s 114, 125, 154). The six extratropical peat

fires are of interest for characterizing extratropical peat fire emissions, but given the high variability of this source, we don't use them here to estimate tropical peat fire emissions. Of the three fires that burned Indonesian peat one of these (#154) was a "room burn" optimized for certain lengthy aerosol experiments, but subject to significant unnatural trace gas losses (Stockwell et al., 2014). Of the two "stack burns" of Indonesian peat, one (#125) had an MCE of 0.872, well above the field range of 0.687-0.835 (this study; Stockwell et al., 2016a), perhaps due to over-drying the

sample. This leaves just fire #114 (MCE 0.744) as ideal for representativeness and supplementing field data. Comprehensive trace gas emissions reported by Stockwell et al. (2015) for this fire are included in Tab. S3.

The 2016 large-scale Fire Influence on Regional and Global Environments Experiment (FIREX, https://csl.noaa.gov/projects/firex/firelab/) Missoula fire lab component also included one stack burn of Indonesian peat (#55). This fire had an MCE (0.831) above our updated field-average MCE (0.76), but lower than four of the

field fire MCEs. EFs from this fire reported in Selimovic et al. (2018) and Koss et al. (2018) are included in Tab. S3. Note the EFs from Koss et al. were scaled up by a factor 1.1394 to reflect the actual fuel carbon fraction (0.5697) rather than the originally assumed fraction (0.50). Fig. 2 also shows the lab burns we have selected as representative in EFCH$_4$ versus MCE space. Both lab fires fall within the field range, but fire #114 is near the top of the CH$_4$ range and fire #55 is near the top of the MCE range. The average EFs from these two lab fires appear to be reasonably

representative and the value of even a small increase in sample size is illustrated.

It is worth noting a subtle difference between lab fire sampling and field sampling. In the lab we measure the total emissions from about 1 kg of peat as it is burned over a 25-40 minute period. The emissions can change dramatically over this time because the ratio of pyrolysis of biomass to gasification of char decreases as uncharred fuel in the limited sample becomes more scarce (Yokelson et al., 1997). This is illustrated in Fig. 6 where the molar ratio of

methanol (a pyrolysis product) to methane (enhanced during gasification) decreases from near 0.3 to about 0.013 (a factor of ~23) as fire #114 consumes a finite sample over 25 minutes. In contrast, in the field we acquire grab samples of a moving fire producing smoke in a mix of fuels at different points along a pyrolysis/gasification trajectory





somewhat like that shown in Fig. 6. As expected, the methanol to methane molar ratio obtained by integrating over whole representative lab fires (0.11 ± 0.04) is similar to the study-average methanol to methane molar ratios measured in the field (0.12 ± 0.01), and both results are near the middle of the range shown in instantaneous values. An assumption we make in this work is that random grab sampling in the field captures the most representative emissions,

but fire-integrated lab results can also be representative of real fires and used for species when no field data is available. In addition, the lab trajectory likely gives some insight into the high variability in field samples. For example, at the plume level Stockwell et al. (2016a) observed methanol to methane molar ratios of 0.127 ± 0.071 (n = 35).

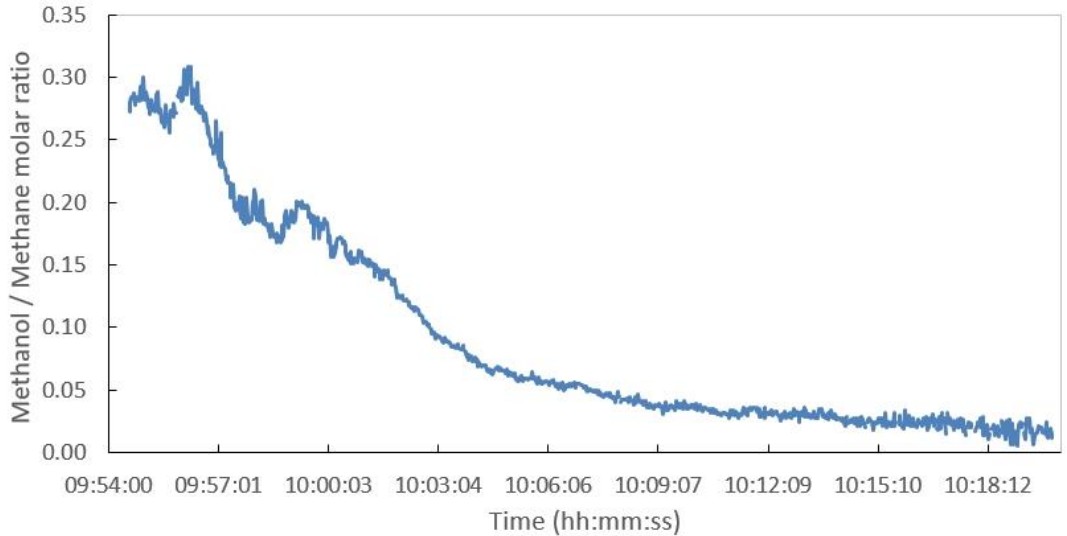

**Figure 6.** Methanol is an indicator of pyrolysis of fresh fuel while methane indicates gasification of charred fuel. The ratio of
methanol to methane drops from near 0.3 to about 0.013 (a factor of ~23) over 25 minutes while burning an approximately one kg sample of Kalimantan peat as fire #114 during FLAME-4 in 2012.

In light of the above discussion, in Tab. S3 we present an extensive set of field average trace gas EFs for tropical peat fires based on sampling by FTIR and WAS at a wide variety of Southeast Asian sites (this work; Stockwell et al.,
2016a; Smith et al., 2018). For $CO_2$, CO, $CH_4$, and MCE we also include the data of Hamada et al. (2013) and Huijnen et al. (2016) in the field average. Tab. S3 also presents lab average EFs computed using FTIR and high-resolution proton-transfer-reaction time-of-flight mass spectrometry (PTR-TOF-MS) data from the two most representative lab fires identified above. The lab average is based on much less sampling than the field, but it significantly expands the amount of measured species due to the broad sensitivity of the high-resolution mass spectrometry technique that was
used in the lab. We note that plotting the lab average EF versus the field average EF for the ~25 species measured in both settings shows good overall agreement (Fig. S1, slope 1.11, $r^2$ 0.854). When methane and ammonia are excluded (both are gasification indicators), agreement between lab and field EFs improves further (slope 1.04, $r^2$ 0.88). Next, we summarize the main features of our newly-computed averages for tropical peat fire emissions.

The three largest trace gas EFs in our new field-average are $CO_2$ (1538 ± 58 g kg$^{-1}$), CO (305 ± 56 g kg$^{-1}$), and $CH_4$
(10.3 ± 2.5 g kg$^{-1}$). These EFs are significantly different from the earlier set of trace gas EFs for tropical peat burning



from a single laboratory peat fire (Christian et al., 2003) that were adopted in IPCC guidelines (Table 2.7 in IPCC, 2014). The reductions for the two main greenhouse gases are $CO_2$ (–10%) and $CH_4$ (–49%). The increase for CO is 45%. Not to be overlooked as a major emission is $H_2$, which is produced in similar amounts to $CH_4$ on a molar basis. Not including nitrogen-containing species, which are discussed separately, but including both lab and field data, the

next largest EFs after $CH_4$ are (units g kg$^{-1}$): acetic acid (4.45), methanol (2.48), ethane (2.00), ethene (1.50), acetaldehyde (1.50), propane (1.38), benzene (1.30), and propene (1.23). When both a field and lab average are available for the same species, we have preferred the field average values for hydrocarbons measured by WAS and all species measured by FTIR, where the latter accounted for much more sampling. In general, we prefer the lab value for oxygenated species measured only by WAS in the field (acetaldehyde in the list above), because of relatively high

uncertainty for WAS OVOCs (Simpson et al., 2011) and for the species where no field data is available. Compared to other biomass fuels, the dominance of acetic acid and the ranking of ethane above ethene stand out for peat fires where the latter observation is consistent with relatively high alkane emissions in general from semi-fossilized biomass. The glycolaldehyde to acetic acid ratio was < 3% for peat, about a factor of ten lower than usual for other biomass fuels, and the peat-fire glyoxal emissions were also low, likely due in both cases to the low cellulose content of peat

(Richards, 1987). For glyoxal only, we replaced the Koss et al. (2018) data with the data from a specific spectroscopic technique that also used a shorter sample line (Zarzana et al., 2018). Other acids emitted include isocyanic acid (HNCO, 0.574 g kg$^{-1}$), formic acid (0.430 g kg$^{-1}$), nitrous acid (HONO, 0.208 g kg$^{-1}$), and methylbenzoic acid (0.127 g kg$^{-1}$). Comparing the sum of methylglyoxal plus acrylic acid measured by Koss et al. (2018) to the specific methylglyoxal measured by Zarzana et al. (2018) suggests that acrylic acid accounts for about one-third of the signal

at that exact mass and has an EF of 0.0537 g kg$^{-1}$ with a remaining 0.106 g kg$^{-1}$ due to methylglyoxal.

The lab average column in Tab. S3 includes data for 25 g kg$^{-1}$ of NMOG not measured in the field of which 7.83 g kg$^{-1}$ is a reasonable estimate of the amount of detected, but unassigned (unknown) NMOG mass. The unknown NMOG mass is primarily high molecular mass oxygenated volatile organic compounds (Stockwell et al., 2015; Hatch et al., 2015; 2017; Koss et al., 2018) and it accounts for roughly 14% of the total NMOG mass. Listing NMOG species in

order of increasing mass in Tab. S3 facilitates compound location since there are often multiple common names. In addition, this format simplifies determining the most abundant isomers when they are not resolved by the mass spectrometers in the lab studies, but are measured by GC in the WAS field samples. While not a direct comparison, reasonable agreement is seen for the mass total and sum of isomers at e.g., $C_4H_8$ (m/z 56, butenes), $C_5H_{10}$ (m/z 70, mainly pentenes and methyl butenes), $C_8H_{10}$ (m/z 106, ethylbenzene and xylenes), and $C_9H_{12}$ (m/z 120, $C_9$ aromatics).

In contrast, the mass total is significantly larger at $C_4H_6$ (m/z 54, butadienes), $C_5H_8$ (m/z 68, isoprene and pentadienes), and $C_{10}H_{16}$ (m/z 136, monoterpenes). When agreement is poor it may be due largely to the presence of unmeasured isomers. A more direct, more in-depth, analysis of isomer speciation addressing over 500 compounds measured by two-dimensional GC is presented for peat fire smoke and other types of biomass burning in Hatch et al. (2015; 2017). Contributing isomers for the PTR-TOF-MS data we show in Tab. S3 were also characterized using GC-PTR-TOF-

MS by Koss et al. (2018).

Turning to nitrogen species, ammonia ($NH_3$ 5.34 g kg$^{-1}$) and hydrogen cyanide (HCN 4.77 g kg$^{-1}$) are by far the two major emissions based on field data. The lower EF $NH_3$ in the lab average (1.81 g kg$^{-1}$) is the largest lab-field



difference for any major species. The lab data are based on open-path FTIR while the larger field values are based on open-path FTIR or specially-coated closed cell FTIR (Stockwell et al., 2016a; Yokelson et al., 2003). Thus, the discrepancy is probably the small sample size in the lab average since the largest reported EF $NH_3$ in the literature is the Christian et al. (2003) lab sample and including it in the lab average would raise it to 7.85 ± 10.46 g kg$^{-1}$. The

molar ratios of HCN or acetonitrile to CO have important applications as biomass burning (BB) tracers and these ratios (0.0162 and 0.00165, respectively) are higher for peat combustion than other types of BB (Crounse et al., 2009; Akagi et al., 2011; Coggon et al., 2016). Acetonitrile has only been measured in the lab for peat fires to date. The lab data also adds EFs for many less abundant nitriles, amines, imines, etc. Acetamide is important as an air toxic and a precursor to another air toxic (isocyanic acid, HNCO, Roberts et al., 2011). Stockwell et al. (2016a) discussed the high

acetamide emissions measured in FLAME-4 (4.2 g kg$^{-1}$) and acetamide atmospheric chemistry in some detail. Recent work on amide atmospheric chemistry is described elsewhere (Zuo et al., 2021; Ni et al., 2021). Adding the FIREX lab data lowers the peat fire acetamide average EF to 2.25 g kg$^{-1}$, but it's still substantial and future field measurements of this compound would be valuable.

The largest EF for a sulfur compound was measured by FTIR for $SO_2$ (3.42 g kg$^{-1}$) in the FIREX lab fire. This

observation used an isolated, but weak, infrared $SO_2$ band and had low signal to noise suggesting an uncertainty of at least 50% (Selimovic et al., 2018). $SO_2$ was not detected by FTIR in the extensive 2015 Central Kalimantan field sampling and only detected by FTIR in one sample of North Carolina, coastal, temperate peat out of the nine global peat samples burned in FLAME-4 (Stockwell et al., 2015), but was emitted at high levels (4.26 g kg$^{-1}$) in that fire. Apparently, $SO_2$ is occasionally a major emission from peat, likely traced to fuel S variability. Another major sulfur-

containing emission measured during the FIREX lab fire was $H_2S$ with an EF of 0.254 g kg$^{-1}$. Both carbonyl sulfide (OCS, 0.14 g kg$^{-1}$) and dimethyl sulfide (DMS, 0.03 g kg$^{-1}$) were consistently emitted by peat fires based on WAS field measurements. The South Sumatra average EFs for these two species were more than twice the overall 2015-2019 field averages, with the South Sumatra EF for OCS even larger than the lab $H_2S$ at 0.356 g kg$^{-1}$. Within South Sumatra the sum of OCS and DMS ratioed to CO ($\Delta OCS+\Delta DMS / \Delta CO$, ppt / ppb) varied by a factor of six (0.18 –

1.16) among sites in the province. Variability was high within all three villages sampled and almost a factor of four at Senasi Mulya (0.298 – 1.16). Large-scale influences on soil S likely include volcanoes and marine sediments (Gras et al., 1999) while fine scale variability could result from the application of agricultural chemicals or manipulation during canal or road building. $\Delta OCS+\Delta DMS / \Delta CO$ was correlated with $\Delta NMHC / \Delta CO$ ($r^2$ 0.88) suggesting that combustion chemistry also influenced the variability. FLAME-4 and FIREX both reported methanethiol (0.04 g kg$^{-1}$), and

thiophene and methyl thiophene (both ~0.03 g kg$^{-1}$) were also observed in FIREX. These three additional lab EFs for sulfur compounds are similar to DMS in magnitude.

Singly-substituted, halogenated methane compounds measured in the field consistently had fairly reproducible EFs with chloromethane (0.157 ± 0.014 g kg$^{-1}$) about a factor of ten higher than iodomethane (0.0157 g kg$^{-1}$), and bromomethane (0.0139 g kg$^{-1}$). The sum of field-measured EFs for S compounds was well correlated with

chloromethane (slope 0.89, $r^2$ 0.82) when excluding South Sumatra, potentially implicating a link to the use of agricultural chemicals. The observed ΣS / chloromethane mass ratio in South Sumatra was higher at ~2.7 and including





it in the above analysis reduced r$^2$ to 0.13. This is consistent with a large, additional, non-agricultural sulfur source in South Sumatra.

Air toxics in peat smoke and some exposure and risk estimates for the 2015 fire season in Palangka Raya were discussed at length in Stockwell et al. (2016a). Here we simply list seven of the major gas phase hazardous air

pollutants (HAPs) that are emitted by peat fires: HCN (4.77 g kg$^{-1}$), formaldehyde (0.818 g kg$^{-1}$), benzene (1.30 g kg$^{-1}$), 1,3-butadiene (0.151 g kg$^{-1}$), acrolein (0.31 g kg$^{-1}$), acetamide (2.25 g kg$^{-1}$), and HNCO (0.574 g kg$^{-1}$). The latter three are based on lab data and a mass spectrometry deployment in the field would be useful for a better assessment. Interpreting BB HAPs emissions in light of recommended exposure limits is also discussed elsewhere (Akagi et al., 2014; O'Dell et al., 2020).

**3.3 PM$_{2.5}$ size distribution, emission factors, chemistry, aging, and optical properties**

In this section we compare the available representative gravimetric measurements of EF PM$_{2.5}$ for tropical peat burning; summarize a few key physical and chemical features, optical properties and aging results; and provide references for further details. We compare only gravimetrically-measured EFs because uncertainty in density and the size-dependent mass scattering efficiency impacts the other available PM emissions estimates that are based on light

scattering. Even a gravimetrically-calibrated, light-scattering PM measurement can be impacted by size distribution changes at the fire source or with smoke aging (Akagi et al., 2012; Carrico et al., 2016; Kleinman et al., 2020). Carrico et al. (2016) show a typical fresh peat smoke size distribution in their Fig. 3d with peak diameter near 100 nm, much smaller than the dust often used to calibrate light-scattering instruments. Fig. 4 in Carrico et al. (2016) shows the initial smoke size distribution evolving from a peak diameter of 95 nm to a peak of 60 nm over ~15 min during lab peat fire

#125 in FLAME-4. At the same time the methanol to methane molar ratio measured by FTIR decreased by a factor of three indicating a decrease in the pyrolysis to gasification ratio (Stockwell et al., 2014). The decrease in pyrolysis / gasification likely contributes to the emission of smaller less scattering particles as also implied for another typical BB fuel in Fig. 5 of Carrico et al. (2016). It's unclear if size changes could impact suggestions that PM mass emissions change with peat fire age in the field (Roulsten et al., 2018). It's also unclear if any fire-age dependence of PM mass

emissions could bias random sampling in the field or how to determine fire age operationally. The concept of fire age has limitations when applied to a field fire moving into fresh fuels. A final remark on size is that the small size of peat smoke particles, along with their low solubility (vide infra), would tend to reduce their efficiency as cloud condensation nuclei (Carrico et al., 2016; Chen et al., 2017). Of the many biomass fuel types burned during FLAME-4, peat was the only fuel that produced no detectable ice-nucleating particles (Levin et al., 2016).

In Fig. 7 we compare gravimetric measurements of EF PM$_{2.5}$ versus MCE from the extensive field measurements of Jayarathne et al. (2018) and the fire-integrated lab measurements from FLAME-4 (fire #114, Jayarathne et al., 2014), Christian et al. (2003), and Watson et al. (2019). The Watson et al. (2019) average EF PM$_{2.5}$ for four lab peat fires is 22.6 ± 3.1 g kg$^{-1}$. The other two lab fires included are more variable at 6 g kg$^{-1}$ (Christian et al., 2003) and 38 g kg$^{-1}$ (Jayarathne et al., 2014), but they average to 22 g kg$^{-1}$. All the lab data taken together average 22.4 ± 10.4 g kg$^{-1}$. This

is ~30% higher than the more extensive field average of 17.3 ± 5.8 g kg$^{-1}$ (Jayarathne et al., 2018). While this difference is not statistically significant, somewhat lower "real" EF in the field could occur from decreased partitioning of organic





gases to organic aerosol (increased evaporation) at the higher field temperatures (33-37 ºC field versus ~15-20 ºC lab) (May et al., 2013; Selimovic et al., 2019; 2020).

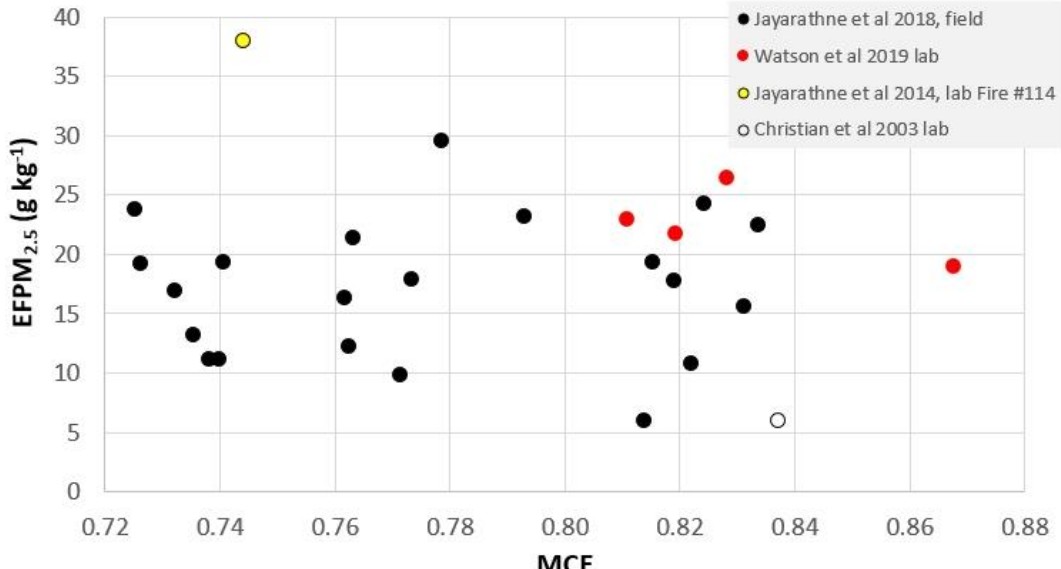

**Figure 7.** Gravimetric determination of EFPM$_{2.5}$ versus MCE for tropical peat fires. See text for details.

Extensive chemical analysis of tropical peat fire PM$_{2.5}$ is provided for the field measurements of Jayarathne et al. (2018). Extensive chemical analysis of laboratory tropical peat fire PM$_{2.5}$ is presented in Jen et al. (2019, FIREX fire #55), and Watson et al. (2019). Detailed chemical analysis of lab tropical peat fire PM$_{10}$ is presented in Iinuma et al. (2007). Here we summarize the main features. Peat fire PM$_{2.5}$ is mainly organic and insoluble. Jayarathne et al. (2018)

10 found that organic carbon (OC) alone accounted for $72 \pm 11\%$ of PM$_{2.5}$ mass (EF OC ~12.5 g kg$^{-1}$) with only 16% of OC being water soluble (WSOC). The low WSOC is consistent with high alkane content and low sugar content (Jayarathne et al., 2018; Jen et al., 2019; Iinuma et al., 2007). The ratio of organic aerosol (OA) to OC was $1.26 \pm 0.04$, which is lower than for other types of fresh BB aerosol (1.4-1.8) due to the semi-fossilized peat fuel and resulting water-insoluble, aliphatic-rich OC.

15 Thermal-optically measured elemental carbon (EC) was $1.1 \pm 0.5\%$ of PM$_{2.5}$ mass (~0.19 g kg$^{-1}$) with low EC expected for smoldering combustion (Jayarathne et al., 2018; Christian et al., 2003; Selimovic et al., 2018). Even lower optically-equivalent black carbon (BC, Li et al., 2019; Bond and Bergstrom, 2006; Subramanian et al., 2007) emissions were reported based on photoacoustic field measurements at 870 nm by Stockwell et al. (2016a). They reported EF BC of $0.0055 \pm 0.0016$ g kg$^{-1}$ (n = 7) and an approximate OC to BC ratio of ~2900. The photoacoustic lab measurement

20 of EF BC for Kalimantan peat (fire #55) by Selimovic et al. (2018) was also low at 0.0026 g kg$^{-1}$ (n = 1). Combining these photoacoustic measurements gives an EF BC of $0.0052 \pm 0.0018$ g kg$^{-1}$ (n = 8). Even this relatively low photo-acoustically-determined EF BC could be an overestimate due to partitioning-driven, coating formation on soot





entrained from the background (Li et al., 2019; May et al., 2013) or the weakly-absorbing microscopic charcoal particles that can naturally occur in smoke plumes even without flaming (Han et al., 2007; 2010).

Water-soluble ions and metals account for a small fraction of $PM_{2.5}$ mass (Jayarathne et al. 2016, Iinuma et al. 2007, Watson et al. 2019). The dominant water-soluble ions measured in the field by Jayarathne et al. (2018) as mg per g of

$PM_{2.5}$ were ammonium (5.1), chloride (4.2), and sulfate (1.4). Metals are often used as tracers in PM source apportionment studies (e.g. Khanum et al., 2021) and 15 metals were quantified in peat fire smoke in the field by Jayarathne et al. (2018), with these metals accounting for < 0.15% of $PM_{2.5}$ mass. The dominant metals in the peat smoke field data were (reported as mg per $gPM_{2.5}$): Cu (0.74), Zn (0.40), and Fe (0.27). These same three metals are of interest for their major role in important neurodegenerative diseases (Ben-Shushan et al., 2021) and other studies

have linked BB smoke metals to neurological hazards (Scieszka et al., 2021). Jayarathne et al. (2018) also reported field-measured values for a large suite of PAHs, alkanes, selected sugars, lignin decomposition products, and sterols. A longer list of metals and data for other $PM_{2.5}$ constituents are provided in Tab. S6 of Watson et al. (2019).

Ahern et al. (2019) aged tropical peat fire smoke in dual smog chambers during FLAME-4 and Watson et al. (2019) aged tropical peat fire smoke in an oxidative flow reactor. Both studies reported insignificant net mass gain resulted

from the combined effects of secondary organic aerosol formation and primary organic aerosol evaporation. However, Chen et al. (2018) reported that the formation of secondary organic aerosol in peatland fire smoke did increase the degree of oxygenation and promote hygroscopicity, and they reviewed related literature.

Stockwell et al. (2016a) performed real-time co-sampling of seven Central Kalimantan peat fires with FTIR and PAX to measure and scale $PM_{1.0}$ optical properties. They reported the SSA and EFs for absorption and scattering at 870 and

401 nm and the AAE. Liu et al. (2014) reported the SSA at 781, 532, and 405 nm and the AAE for the representative FLAME-4 lab fire #114. For the same lab fire, Pokhrel et al. (2016) reported the SSA at 660, 532, and 405 nm and the AAE. In Fig. 8a we plot the field and lab data for initial peat-smoke SSA versus MCE. Consistent with low BC emissions, the near-IR and visible SSA is always close to one regardless of wavelength or MCE with an average visible initial SSA that is based on all the lab and field data of 0.998 at the field average MCE of 0.76. The

measurements of peat smoke optical properties cited above were made on dried aerosol. We are not aware of measurements of particle growth and scattering increases at high humidity for pure peat smoke (f(RH), Gras et al., (1999)), but the growth may be small for pure, fresh peat smoke due to the above-mentioned low hygroscopicity.

During late October 1997, as part of the Pacific Atmospheric Chemistry Experiment 5 (PACE-5) campaign, airborne sampling of peatland fire smoke/regional haze was conducted in coastal South Kalimantan during an intense El Niño

event (Sawa et al., 1999; Stockwell et al., 2016a). Gras et al. (1999) estimated the SSA for the 1997 Kalimantan regional smoke as 0.98, which implies a modest contribution from non-peat BB fuels since they tend to burn with more flaming and BC emissions and lower SSA (0.7 – 0.96, Christian et al., 2003; Reid et al., 2005; Liu et al., 2014; Pokhrel et al., 2016; Selimovic et al., 2018). During the 2015 intense El Niño event Eck et al. (2019) measured a visible SSA of 0.975 for the Palangka Raya AERONET site in Central Kalimantan that was inundated with fairly fresh

smoke and estimated that 80-85% of regional smoke was from burning peat. For source apportionment purposes it should be kept in mind that several hours of smoke aging usually increases the SSA (Abel et al., 2003; Yokelson et al., 2009; Kleinman et al., 2020).





In contrast to the minimally-varying fresh peat smoke visible SSA, Fig. 8a also shows that the SSA at 405 or 401 nm has an MCE dependence, a finding consistent with the previously noted tendency for higher emissions of brown carbon (BrC) at lower MCE (Liu et al., 2014; Selimovic et al., 2018). Based on a fit of all the lab (n = 2) and field (n = 7) measurements, the 405-401 nm SSA is 0.958 at the field-average MCE (0.76) for tropical peat fires. In Fig. 8b, a

5  similar analysis suggests an AAE of 5.7 at the field average MCE. This AAE indicates that about 97% of absorption at 401-405 nm is due to brown carbon (Lack and Langridge, 2013). Field measurements of BB smoke have usually reported AAE decreases over the course of hours to days (Selimovic et al., 2020 and references therein).

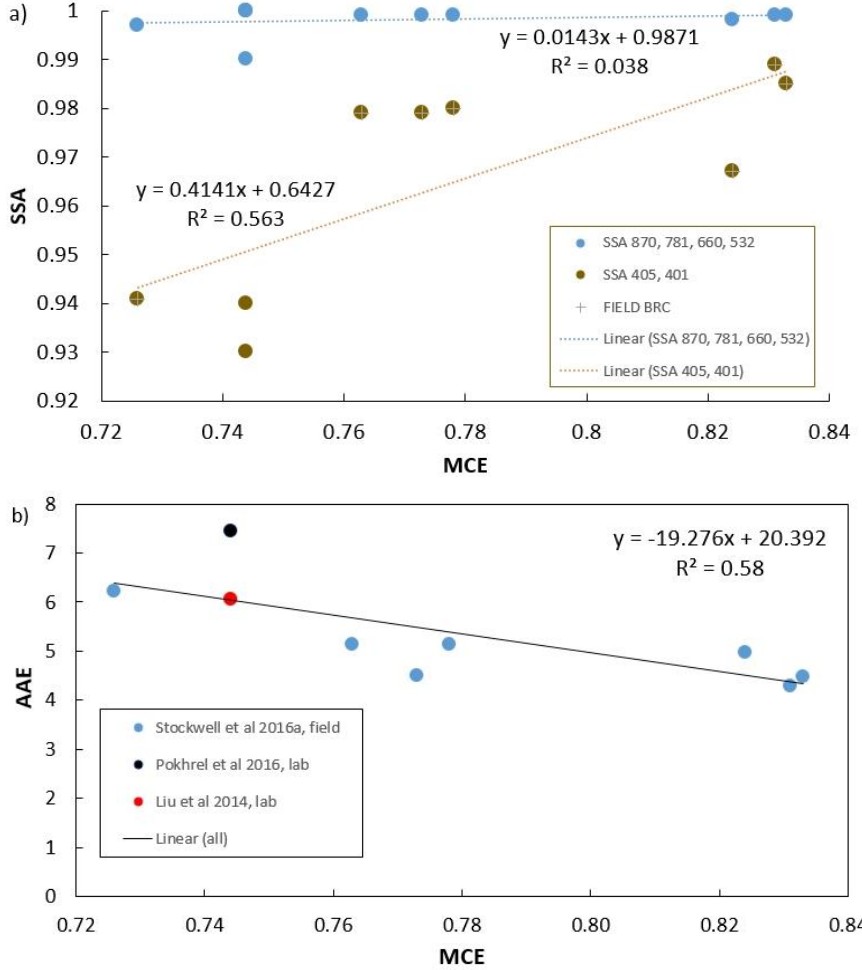

**Figure 8.** a) SSA versus MCE measured in the range 870 – 532 nm (blue). The lab data in this wavelength range (Liu et al., 2014; Pokhrel et al., 2016) is at MCE 0.744 and a value of one is reported in three cases. SSA versus MCE measured at 405 or 401 nm (brown). + indicates field data from Stockwell et al. (2016a). The visible through near-IR SSA near one is consistent with extremely small BC content. More UV absorption by BrC is seen at lower MCE. b) The AAE versus MCE with equation used to calculate an average peat fire AAE at the field average MCE of 0.76.



### 3.4 Context and guidance for using peat fire emission factors

Peat and peatland are not the same thing. Peat is only one component of the total peatland biomass fuel, but it is often the largest component, especially as the dry season progresses and the water table drops. Carbon-14 analysis of PM that impacted Singapore during the 2015 El Nino suggested that 85% of PM was due to burning "ancient" peat
(Wiggins et al., 2018), in good agreement with the Kalimantan SSA-based estimate of the peat fraction of regional biomass fuel consumption by Eck et al. (2019) mentioned above and the estimate of ~83% by Kaiser et al. (2016). The EFs and optical properties in this study are for the initial emissions from burning tropical peat. EFs, fuel consumption, and other properties for burning the other co-located, "above-ground" biomass fuels can be found elsewhere (Reid et al., 2005; Akagi et al., 2011; Liu et al., 2014; Pokhrel et al., 2016; Stockwell et al., 2016b; Andreae
2019; Volkova et al., 2021). Extensive data are available for some above-ground fuels (e.g., tropical forest and crop residue fires), but smoke characteristics for some other above-ground fuels are not measured yet (e.g. ferns). Note that land use/cover can change quickly in the tropics (Miettinen et al., 2016). Next, we discuss some complications and uncertainties regarding the use of the peat fire EFs in this study.

In a bottom up approach EFs are multiplied by a total fuel consumption to generate total emissions for a desired region
and time (Seiler and Crutzen, 1980). The mass of fuel consumed for peat is most often estimated from area burned multiplied by depth burned multiplied by peat bulk density. A mass burned estimated this way can be directly paired with EFs (g kg$^{-1}$) to calculate total emissions. Our 2019 emissions measurements, synthesized with other work, enable more robust and comprehensive regional average EFs to be calculated for burning peat. However, it should be kept in mind that for a single fire or group of fires burning peat, EFs could commonly vary by at least 5% for $CO_2$, 20% for
CO, 25% for $CH_4$, a factor of three for NMOGs and PM, and more for sulfur compounds.

Burned areas, especially small burned areas in SE Asia, are known to be difficult to detect due to high regional cloud cover; orbital gaps; rapid growth of new vegetation, which is strongly associated with shallow burn depth (Cypert, 1961; Kotze, 2013); and other factors (Atwood et al., 2016; Liu et al., 2020; Reddington et al., 2016; Reid et al., 2013; Vetrita et al., 2021). An additional complication with peat fires is that the peat, if present, doesn't ignite everywhere
that the surface fire burns (Graham et al. in preparation). Typically, fine (small diameter, e.g. litter, grass) above-ground fuels must burn to ignite medium above-ground fuels, which then ignite the heavy above-ground fuels (e.g. logs) that can sustain combustion long enough to initiate a sustainable peat fire. Land cover types prone to surface fires that include burning large fuels anecdotally seem more likely to ignite peat, but this has not been studied quantitatively to our knowledge. In any case, a fire's total area is almost always significantly larger than the underlying
peat fire area. To some extent, the underestimate of surface fires can cancel the error associated with assuming the peat ignites under the whole surface fire area. Synthetic aperture radar (SAR) can detect areas of burned peatland, if the surface is dry, even under clouds or thick smoke, and has potential to improve estimates of peat burned (Vetrita et al., 2021).

Depth burned is typically highly variable across any peat fire and hard to measure (Ballhorn et al., 2009). Depth burned
tends to increase strongly with months into the dry season as drought causes the water table to drop (Shawki et al., 2017; Sinclair et al., 2020; Grahame et al. in preparation). For instance, Graham et al. observed seven times greater peat burn depth in September than in August as the dry season progressed. Depth burned tends to decrease when


successive fires burn in the same location (Konecny et al., 2016). Average burn depths are normally reported as an average for the peat fire area and in theory should not be applied to the whole surface fire area. However, using too large a depth and/or applying it to too large an area may cancel the tendency to underestimate surface fire area. Finally, much of the peat may burn well after the combustion of the surface fuels that leads to a fire detection is complete. Peat

fires progress as glowing fronts that spread slowly laterally and downward, i.e. depth increases with time since ignition at a given point. Tropical peat has been reported to burn for up to 20 days on a site (Roulston et al., 2018), long after the surface vegetation was consumed. The duration of peat consumption can vary with fuel moisture, wind, etc. The bulk density of peat can vary from 0.08 to 0.16 g cm$^{-2}$ and was found to be higher for areas that burn more often, which offsets the finding that these previously burned areas also tended to burn less deep (Sinclair et al., 2020). In

light of the above discussion, we note that studies by Kiely et al. (2019; 2020) assume a peat fire to surface fire area of 40% and find improved model performance when assuming burn depth scales with soil moisture as is also done in van der Werf et al. (2010).

Top-down approaches typically involve, as one example, estimating a flux of smoke PM using modeled meteorological fields, assumed plume rise, and a remotely-sensed surrogate for PM (e.g., aerosol optical depth (AOD))

and then computing the ratio of PM produced to remotely-sensed fire radiative power (FRP) to derive emission coefficients (ECs) that can be applied to hotspots without knowledge of fuel consumption (Lu et al., 2019). Alternatively, FRP may be used to infer mass consumption of fuel and obtain emissions of interest using EFs. When ECs cannot be measured for a species they can be estimated from EF ratios. Sources of uncertainty include missing hotspots (especially problematic for peat combustion) and/or missing smoke due to clouds, cloud mask, orbital gaps,

and small or thin plumes; uncertain windspeed; and evolving or uncertain ratios between ECs or EFs or of PM to AOD or gases (Lu et al., 2021; Shi et al., 2019; Wooster et al., 2021). In summary, depending on the bottom-up or top-down approach employed, missed fires, unknown fuels, and other issues can be important, but the larger more robust suite of EFs for trace gases and PM$_{2.5}$ we present here should help reduce overall uncertainty.

## 4 Conclusions

With the completion of this study, authentic "fires of opportunity" burning in southeast Asian tropical peat deposits have now been sampled in the field over a broad range of climate conditions and geographic locations. Combined with earlier field sampling of burning peat in SE Asia, we now have more robust field-based knowledge of the average emissions and the natural variability for EF PM$_{2.5}$ (17.3 ± 5.8 g kg$^{-1}$), PM$_{2.5}$ chemistry, dry aerosol optical properties, and ~90 trace gas EFs including HAPs and the major GHGs. Adjustments to IPCC recommended EFs for peat burning

are supported as follows: CO$_2$ (-10% to 1538 g kg$^{-1}$), CO (+45% to 305.1 g kg$^{-1}$), and CH$_4$ (-49% to 10.3 g kg$^{-1}$). Many (i.e. more than a factor of ten) other EFs have been added or changed significantly since the 2003 study of a single sample used by IPCC, e.g. EF NH$_3$ decreased from 19.9 to 5.34 g kg$^{-1}$. Further benefits could result from deploying broadband aerosol absorption and advanced mass spectrometric techniques in the field. For the time being, we have used our improved field characterization as criteria to select the most representative data from parallel, intensive lab

measurements of burning tropical peat that included advanced MS and other powerful techniques. We then combined the selected lab data with the field data to develop a more extensive body of recommended EFs for 230 gases and





numerous aerosol constituents and recommended aerosol optical properties. The complete results are presented in the supplemental tables or cited literature with highlights presented in the text and abstract. We note that the use of multiple techniques and platforms was critical in providing broad characterization. Lab-based MS made it possible to increase the mass of quantified NMOG by about 82% (an additional 25 g kg$^{-1}$) and to estimate that about 86% of the

total NMOG emissions detected in a full PTR-TOF-MS mass scan can currently be named and quantified. MS is the only source of data for some important HAPs such as acrolein (0.31 g kg$^{-1}$), HNCO (0.574 g kg$^{-1}$), and acetonitrile (0.735 g kg$^{-1}$), which is also a BB tracer. The GC analysis of field WAS samples quantified the main GHG emissions and the large emissions of H$_2$ (1.22 g kg$^{-1}$) and alkanes (5.6 g kg$^{-1}$), where the latter are more substantial for peat than other BB types. WAS also contributed detailed speciation of the hydrocarbon isomers at several exact masses and was

the most convenient way to explore regional variability in the field. FTIR in both the field and lab provided an additional overview of the main GHG emissions; overlap with both MS and WAS; and quantification of HCN (4.77 g kg$^{-1}$, a BB tracer), SO$_2$, NO$_x$ (0.31 g kg$^{-1}$), formaldehyde (0.82 g kg$^{-1}$), some sticky/reactive species such as ammonia (5.34 g kg$^{-1}$) and HCl (0.035 g kg$^{-1}$), the major emission acetic acid (4.45 g kg$^{-1}$) and glycolaldehyde (0.11 g kg$^{-1}$), which appears at the same exact mass, etc. FTIR measurements in series after PM collection on filters in the field

enabled off-line quantification of numerous particle constituents as emission factors (Jayarathne et al., 2018). The ability of the compact PAX systems to measure both absorption and scattering in smoke from off-road, burning peat deposits supported recommendations for low BC emissions (0.0052 ± 0.0018 g kg$^{-1}$), high BrC emissions (~97% of absorption at ~401-405 nm), SSA as a function of wavelength, and AAE (5.7). Field and lab experiments consistently measured organic-dominated, mostly insoluble initial PM and multiple lab experiments measured minimal post-

emission OA net mass gain with aging of "pure" peat smoke.

The main application of these new data is to improve estimates of the initial emissions and smoke properties from the substantial peat component of peatland fires. Updated guidance for using the data is provided. Similar data for the initial smoke from some major peatland surface fuel types such as crop residue and tropical forests is available elsewhere (Akagi et al., 2011; Andreae 2019; Liu et al., 2014; Pokhrel et al., 2016; Stockwell et al., 2016b). The

emissions from some surface fuels unique to SE Asia such as ferns still need better characterization. In addition, an airborne campaign is strongly needed in this region to characterize the initial smoke from representative landscape fuel mixtures, evolution of typical mixed-fuel smoke, peatland-fire smoke interactions with urban and biogenic emissions, and the general properties of multisource regional haze.

**Data availability.** The raw WAS data and calculations used in this paper along with detailed site notes, photographs, and maps can be found online (at https://tinyurl.com/yc6yhvx7).

**Author contributions.** All the authors contributed to designing, planning, or executing some aspect of the field measurements. DRB, SM, and IJS measured WAS samples and checked raw data. RJY calculated EFs and wrote the first draft. All authors contributed comments. MAC was additionally responsible for overall project management.

**Competing interests.** The authors have no competing interests.



**Acknowledgements.** Purchase and preparation of the PAXs was supported by NSF grant AGS-1349976 to RJY This work would not have been possible without the excellent support provided by the BOS office in Palangka Raya; notably Grahame Applegate, and the BOSF field team.

**Financial support.** This research was primarily supported by NASA Grant NNX13AP46G to UMCES and UM. The research was also supported by NASA grant NNX14AP45G to UM.

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
