# Peer review of "Tropical peat fire emissions: 2019 field measurements in Sumatra and Borneo and synthesis with previous studies"

_Atmospheric Chemistry and Physics, 2022_

## Author Comment (AC1)

Response to Referee #1

We thank the Referee for their encouraging assessment and constructive suggestions, which improved the paper. The Referee comments are reproduced below (in black) followed by our detailed response (in blue). Page and line number specifications refer to the posted discussion version.

Anonymous Referee #1, 22 Apr 2022

This manuscript reports new emission factors for CO2, CO, CH4, and wide range of organic gases for burning peat. The emission measurements were obtained during 2019 field measurements in Indonesia. The authors combine these recent measurements with previous field measurements and laboratory studies to provide a comprehensive emission factor database for burning Indonesian peat, an important source of regional air pollution. The study provides significant updates to emission factors for CO2, CO, and CH4 and these should have an important impact on peat fire emissions inventories. The methods employed are sound, the paper is well written, and the presentation is clean and concise. The discussion provides valuable guidance for applying the papers emissions factors for peat fire emission inventory development. I have only a handful of comments for the authors to address.

**Specific Comments**

**R1. Discussion**

In the discussion the authors fail to include the Indonesian peat fire EFs reported by Wooster et al. (2018). This study reports in-situ measurements of fresh emissions from pure sub-surface peat fires (as in the current study) and surface fuels + peat fires from fires Central Kalimantan, Indonesia during October 2015 (described as the 'peak' of peat fires). The authors should include Wooster et al. EFs for CO2, CO, CH4, and PM2.5 for pure peat fires. The Wooster et al. EFs for fires involving peat + surface fuels would also be very useful for informing the discussion of applying EFs for bottom-up estimations of emissions from peat fires in SE Asia. While Wooster et al. uses optical methods to report EFPM2.5, they are based on calibration versus gravimetric PM2.5 using simultaneously collected filter samples from the field. Interestingly, their EFPM2.5 is roughly the same as that reported in current study. Also, since they did not measure NMOGs, their EF will be inflated somewhat. Nonetheless, these previous results should be included in the discussion with appropriate qualifications.

**Author's response:** This recommendation strengthened our paper, but it's complicated as we explain next. In an initial skim of Wooster et al. (2018) (abbrev as "W18") we noted the statement in section 3.4: "Huijnen et al. (2012) [12] reported gaseous EFs for a subset of the locations used herein." We take this to mean that the W18 paper supersedes (overrides, updates) the Huijnen et al. (2016) (abbrev as "H16") paper we had already cited. To avoid repeating the same data twice, consistency, and to extract the best "peat only" data we first reread H16. We relocated the source of the data we cited on page 3 of H16 quoted next: "Our mean EFs for these essentially 'peat-only' fires are $255 \pm 39$ g kg$^{-1}$ DM, $1594 \pm 61$ g kg$^{-1}$ DM and $7.4 \pm 2.3$ g kg$^{-1}$ DM, for CO, CO$_2$ and CH$_4$, respectively, see methods. … The uncertainties in the derived EFs are mainly driven by an estimated 10% uncertainty in the carbon content of the burning peat

(~55% for peat-only fires)." The peat-only data in H16 is evidently based on locations 1, 2, and 4 of H16 and the C content is taken from a separate study of peat C that H16 cites by Shimada et al.

Now back to W18. W18 includes new data and also reanalysis of previous data. In more detail, H16 was based only on laser absorption and W18 added OP-FTIR, optically-measured PM2.5, peat C measurements, and featured 5 locations. EFs were calculated without considering the C in the NMOGs as the Referee points out and also, the C in PM was not considered, but we can overlook that as we did for the Hamada et al. study. Why there are two EF for PM2.5 for each fire, but only one EF for gases and if W18 prefers one of the two EF PM values was unclear. We just averaged the two EFs. In W18 the location numbering scheme may or may not be the same. Location 3 is still forest and flaming, but now location 4 is also flaming. Location 5 is potentially an added or renumbered site. From the standpoint of isolating pure peat emissions (our purpose), Table 1 of W18 and the text explicitly mention a non-peat contribution to the emissions at locations 1-4, but not at location 5. Thus we fairly confidently target location 5 for further consideration for deriving true peat only EFs. Table 2 of W18 gives the peat C content for different depths at location 5 as 571 and 585 g/kg, which corresponds to an average C content of 57.8%. Table 4 of W18 reports EFs based on a study-average %C of 61%, which is high compared to other studies and, in any case, not the location 5 measured value. So we opted to normalize the location 5 EFs by 578/610 to arrive at location-5-specific EFs. Also using the average of the two $EFPM_{2.5}$ values we finally arrive at "location 5 peat only" EFs for $CO_2$, CO, $CH_4$, and $PM_{2.5}$ of 1623, 306.7, 4.95, and 12.52 g/kg, respectively. These EFs are within the range of other studies. Also on the positive side, the FTIR and laser data from this study agree well with each other and the MCE for location 5 (0.771) is close to our study average for pure peat.

To incorporate this process into our paper we assumed that W18 supersedes H16. We deleted all mention of H16 from the text, tables, figures, and references. We replaced the H16 data with the W18 data and computed new averages and updated all the relevant text, tables, and figures.

We also added text on page 9 explaining that we include the W18 data from location 5 for pure peat, after normalizing the reported EFs to the location 5 peat %C.

Page 9, lines 6-9:
Old text: "Hamada et al. (2013) and Huijnen et al. (2016) each report a study-average MCE and EF $CH_4$ based on limited sampling of peat fires in Central Kalimantan in 2009 and 2015, respectively."
New text: "Hamada et al. (2013) and Wooster et al. (2018) each report MCE and EF $CH_4$ data based on limited sampling of peat fires in Central Kalimantan in 2009 and 2015, respectively. We calculated MCE and EFs for pure peat fires from the data provided for location 5 in Wooster et al. (2018) as explained in detail elsewhere (cite response). The values from these studies …"

Page 13, lines 15-16:
old text "For $CO_2$, CO, $CH_4$, and MCE we also include the data of Hamada et al. (2013) and Huijnen et al. (2016) in the field average."

new text "For $CO_2$, CO, $CH_4$, and MCE we also include data from Hamada et al. (2013) and Wooster et al. (2018) in the field average."

We added our best guess at the W18 pure-peat EFPM$_{2.5}$ on page 17, line 2:
"Wooster et al., (2018) reported a gravimetrically-calibrated optical field measurement of EF PM$_{2.5}$ for their peat only location near 12.5 g kg$^{-1}$, which is also below the lab average."

Regarding the EFs in W18 for peatland fires that include multiple fuels and both flaming and smoldering combustion. This topic is also complicated and a full treatment is beyond the scope of this paper. We do note that representative measurements of fires that include flaming can be difficult with ground-based platforms since much of emissions may be directed at high velocity upward from the flame front where sampling is dangerous (Akagi et al., 2014). Also the choice of what %C to use is tricky when fires are subsampled while consuming a dynamic mix of diverse fuels. A few of the results exhibit unusual trends, probably due to the small sample size rather than measurement errors, but more sampling may be needed to support strong recommendations. For instance: the fires burning a mix of peat and overstory have a higher reported %C than the pure peat. The plots of EF vs MCE for locations 1-5 diverge from the trends usually seen for landscape-scale fires that include both flaming and smoldering. EF $CH_4$ is not correlated with MCE as is usually the case. EF PM$_{2.5}$ is highly correlated with MCE, but has the opposite of the normal dependence: i.e. there is far more PM from flaming in W18. Fig. 6 shows some PM$_{2.5}$ data being collected at extremely high concentrations (e.g. 17-25 mg/m$^3$) that might encourage gas to particle partitioning (May et al., 2013; Selimovic et al., 2020) and could inflate some EFPM. This doesn't seem to have impacted their "pure peat results" from "location 5", but may be why they got some EFPM for burning surface vegetation that were in their words "among the highest ever recorded" and well above the literature average for various surface vegetation types. We reiterate this is not a criticism of the W18 values, but a rationale for encouraging additional measurements to underpin average EFs for mixed fuel fires in Indonesia's exceptionally diverse fire theatre as we stress in the conclusions. We agree W18 illustrates a good example of measuring total emissions rather than trying to compute peat and surface fuel emissions separately and then adding them together. We add a sentence to this effect in Sect. 3.4., page 20, line 11: "The combined emissions from burning both peat and above-ground fuels have been measured, but only for a few fires and just four species (Wooster et al., 2018)."

May, A. A., E. J. T. Levin, C. J. Hennigan, I. Riipinen, T. Lee, J. L. Collett Jr., J. L. Jimenez, S. M. Kreidenweis, and A. L. Robinson: Gas-particle partitioning of primary organic aerosol emissions: 3. Biomass burning, J. Geophys. Res. Atmos., 118, 11,327–11,338, doi:10.1002/jgrd.50828, 2013.

Regarding regional air quality. We agree W18 has a great discussion of the AQ impacts from the peak of a historic fire event and with a rare and much-needed focus on the most heavily-impacted sites, which are in Indonesia rather than Singapore where most previous studies focused. We added W18 to the list of regional air quality studies cited in the introduction.

**R2. Application of EFs for estimating emissions from SE Asia peat fires**

The emissions literature indicates the carbon content of Indonesian peat varies by 30% (44% – 61%, Iinuma et al. 2007, Wooster et al. 2018). This variability is similar to the uncertainties in EF for CO2, CO, CH4, and PM2.5 and may be worth mentioning for those seeking to apply emissions factors.

**Author's response:** We agree that peat carbon content varies. However, papers usually don't present enough detail on how %C is measured for us to rigorously estimate true variability in C content. For example, we know that peat formation should lead to higher C content in peat than in vegetation, yet some studies report peat %C near or below the low end for vegetation in some peat samples. For example 44% and 19% in Iinuma et al. (2007) and Watson et al. (2019), respectively. In our experience, one possible cause of this is that mineral soil or other non-flammable inorganic material is included in the peat sample. Inorganic material doesn't burn or contribute to fuel consumption. The classic approach that we use to offset this is to report the "ash-free %C" (Susott et al., 1996). For example, if 50% of the bulk material is C, but 5% of the sample mass remains as "ash" (nonflammable residue) after combustion, the %C of the burnable fuel is 50/0.95 or 52.6%. Also, several methods of elemental analysis exist, each with their own pros and cons, and the method is not always identified in papers. Finally, for diverse ecosystems, it's best to grind a large sample and then mix before acquiring the typically small subsample actually analyzed. While we can't address this topic in full, thanks to the Referee, we have improved the paper by noting that we report ash-free %C and why on page 7 line 27. It's good to get more recognition of these issues in the literature.

Old text: "We assumed a carbon fraction (0.579 ± 0.025) measured earlier as the average of seven samples of Kalimantan peat (ALS Analytics, Tucson) (Stockwell et al., 2014)."

New text: "We assumed an ash-free carbon fraction (0.579 ± 0.025) measured earlier as the average of seven samples of Kalimantan peat (ALS Analytics, Tucson) (Stockwell et al., 2014). The ash-free carbon content corrects for the potential inclusion of non-flammable inorganic material (e.g., mineral soil) in peat samples."

Susott, R. A., G. J. Olbu, S. P. Baker, D. E. Ward, J. B. Kauffman, and R. Shea, Carbon, hydrogen, nitrogen, and thermogravimetric analysis of tropical ecosystem biomass, in Biomass Burning and Global Change, edited by J. S. Levine, pp. 350-360, MIT Press, Cambridge, Mass, 1996.

The discussion of large-scale emissions estimates for peat burning in SE Asia and the impact of updated EF that is presented in Wooster et al. could inform the discussion in 3.4 Context and guidance for using peat fire emission factors, at a minimum it should be mentioned.

**Author's response:** As noted above, we now highlight the important, alternate approach of measuring total peatland fire EFs in Sect 3.4

**R3.** L10-12 P11: "Compared to other biomass fuels, the dominance of acetic acid and the ranking of ethane above ethene stand out for peat fires where the latter observation is consistent with relatively high alkane emissions in general from semi-fossilized biomass."

Please provide citation or explanation.

**Author's response:** We found this text on page 14. For all the vegetation fuels in Akagi et al. (2011) and Andreae (2019), except peat, ethene is greater than ethane; up to a factor of ~2. Our lab and field EFs for acetic acid in this study range from ~4-5 g/kg, which is also the top of the range for other fire types in Akagi et al. (2011) and Andreae (2019). We added citations to these already-used references to the sentence.

**R4.** Can the author offer any comments on the large difference in EFacetamide for FIREX (0.3 g/kg) and FLAME-IV (4.2 g/kg)?

**Author's response:** The 2012 experiment reported higher acetamide than 2016: 4.21 and 0.292 g/kg, respectively. This is the biggest difference by far for any species measured on both PTR-TOF-MS. In addition to the high natural variability that is seen in the EFs for easily measured species consisting of only C and H like methane, part of this could be due to the larger variation in fuel N than fuel C. In fact, there was higher fuel N in 2012 (2.57%) compared to 2016 (1.57%). However, acetamide may depend more on actual precursor compounds, which could vary for example with degree of decomposition, etc., rather than total N and neither experiment had info beyond total N. Another factor could be experimental. Every PTR-based instrument will have a different sensitivity for acetamide that depends on numerous factors including possible reagent ion depletion at high sample concentrations, collision energies in the reaction chamber, tuning of the fields that guide ions, etc. Early studies mostly used calculated sensitivities, but the trend is to calibrate the sensitivity for more and more species. However, calibration also presents challenges and will not be possible ultimately for all the hundreds of mass peaks, many of which are due to a dynamic mix of isomers. In this context, the "high" 2012 EF was based on a calculated sensitivity and the "low" 2016 EF was calibrated. The difference in assumed sensitivity by the two approaches is unusually large, about a factor ten lower for 2012. Clearly an underestimated sensitivity could inflate the 2012 EF. Because the 2016 data was calibrated it is probably more accurate, but we can't be sure by how much since the 2012 instrument was not both calibrated and compared to a calculated sensitivity. On the other hand, the 2012 instrument had a shorter, warmer sample line that might have better transmitted any compounds that might be extremely sticky. Other possible measurement uncertainty could involve potentially varying degrees of fragmentation of higher masses or potentially varying interference. The assignment of this mass peak to acetamide is based on its known presence in tobacco smoke and other indirect evidence as discussed in the original papers. However, there are other C2H5NO compounds shown on the Chem Spider website. Acetamide is the most likely but other structures could conceivably contribute. Finally, we note results from other PTR-based BB studies. Permar et al used a calculated PTR sensitivity and report an EF for acetamide for western wildfires of 0.04 ± 0.012 g/kg or about 7 times lower than our "low" 2016 peat value. Yokelson et al (2013) used a calculated sensitivity and report similar low EF from 0.06 to 0.09 g/kg for generic forest fuel at the acetamide mass (59), but a much higher EF of 1.2 g/kg (about 4 times our 2016 peat value) for organic soil, which is a peat precursor. Thus, peat fires seem to emit much higher acetamide than generic BB, but more sampling is needed, potentially with other techniques. Unlike the gravimetric vs optical issue for PM we don't have other data from an accepted reference method for acetamide for tropical peat.

We updated the text as follows:

Old sentence "Adding the FIREX lab data lowers the peat fire acetamide average EF to 2.25 g kg$^{-1}$, but it's still substantial and future field measurements of this compound would be valuable."

New text: "Adding the FIREX lab data lowers the peat fire acetamide average EF to 2.25 g kg$^{-1}$. The lower FIREX value is likely more accurate based on improved calibration, but part of the difference likely reflects the lower fuel N in FIREX than FLAME-4, 1.57 and 2.57%, respectively. In any case, emissions of acetamide from peat and organic soil fires appear to be much larger than from burning above-ground biomass fuels (Permar et al., 2021; Yokelson et al., 2013) and future field measurements of this compound, potentially incorporating additional techniques, would be valuable."

Permar, W., Wang, Q., Selimovic, V., Wielgasz, C., Yokelson, R. J., Hornbrook, R. S., Hills, A. J., Apel, E. C., Ku, I-T., Zhou, Y., Sive, B. C., Sullivan, A. P., Collett Jr, J. L., Campos, T. L., Palm, B. B., Peng, Q., Thornton, J. A., Garofalo, L. A., Farmer, D. K., Kreidenweis, S. M., Levin, E. J. T., DeMott, P. J., Flocke, F., Fischer, E. V., and Hu L.: Emissions of trace organic gases from western U.S. wildfires based on WE- CAN aircraft measurements, J. Geophys. Res., 126, e2020JD033838. https://doi.org/10.1029/2020JD033838, 2021.

Finally, instrument/study discrepancies/disagreements, even for a single species, can be a major topic that persists for years. We prefer to limit this paper to reporting best efforts in representative smoke. However, we've already highlighted the paper describing a detailed comparison of most of the instruments involved under nearly ideal lab conditions by Hatch et al. (2017) on page 14 line 32. The cited papers we source data from also discuss accuracy at length.

**R5.** Technical – Typo? Table1: a couple entries with n=2 bur R2!= 1

**Author's response:** We don't force the intercept on these plots so 2 points determine a line and $r^2$ is one. The choice of forcing typically has no significant impact. For instance, for the two 8 October samples from Temperai South Sumatra the unforced slope and $r^2$ are 13.300 and 1.000, while forcing gives 13.303 and 0.999999862. For the two Senasi Mulya South Sumatra samples on 9 November unforced gives 26.712, 1.000, while forced gives 26.651, 0.9999898. The ratio of unforced to forced slopes for these cases are 0.99974 and 1.002289.

**References**

Iinuma, Y. et al. (2007) Source characterization of biomass burning particles: The combustion of selected European conifers, African hardwood, savanna grass, and German and Indonesian peat, J. Geophys. Res.-Atmos., 112, D08209, https://doi.org/10.129/2006JD007120

Wooster et al. (2018) New tropical peatland gas and particulate emissions factors indicate 2015 Indonesian fires released far more particulate matter (but less methane) than current inventories imply, Remote Sensing,10, 1–31, https://doi.org/10.3390/rs10040495, 2018.

---

## Author Comment (AC2)

Response to Referee #2

We thank the Referee for their encouraging assessment and constructive suggestions, which improved the paper. The Referee comments are reproduced below (in black) followed by our detailed response (in blue or red). Page and line number specifications refer to the posted discussion version.

Anonymous Referee #2, 13 Jun 2022

The manuscript presented by Yokelson et al. reports on emission factors of a wide range of species obtained from multiple fires at different location in Indonesia in 2019. It is well written, the methodology is well explained, and the results are discussed in a sufficient manner. The updated emission factors that this work provides is greatly needed to address the impact of tropical peat fires on the atmospheric composition on regional and global scales. Especially the discussion on the usage of the derived dataset for fire emission inventories is very helpful and will allow to further constrain the influence of these fires on air quality and climate. I suggest the publication of this work in ACP once my comments are addressed by the authors.

Specific comments:

1. P18, L1-12: In addition to the potential impact of metals for neurodegenerative diseases, iron is important to marine biota potentially leading to biogeochemical feedbacks. Considering the exceptional strength that Indonesian peat fires can have, could you comment on the potential importance of the emitted iron on the regional ocean fertilization? Compared to other iron sources in this region, can you estimate the relative importance of these fires?

**Authors response:** Thank you for this very interesting comment. We briefly summarize a very large body of work. Ocean dynamics, deposition, sediments, etc. can supply Fe to the ocean's near-surface waters. Dust is the main source of total atmospheric Fe, but dust Fe has low solubility and, usually, slowly increasing solubility with aging. In contrast, pyrogenic Fe (e.g., Fe in aerosol from smelting, combustion of biomass, coal, and liquid fuels) can have high initial solubility and/or rapidly increase in solubility with aging. Thus, pyrogenic Fe has been suggested to be the major source, of deposited and then dissolved oceanic Fe over large regions including Indonesia (Fig 6 in both Conway et al., 2019 and Ito et al., 2019). Dissolved Fe can increase ocean productivity and, if some of the "extra carbon" is exported to the deep ocean, also counteract global warming. However, actual deep ocean C-sequestration in response to deposition of atmospheric Fe may have low significance in some equatorial waters compared to the effects of ocean dynamics (Winckler et al., 2016). It would require isotope data or a specialized model that includes a regionally-appropriate aging scheme to estimate the relative contribution to dissolved Fe from BB compared to other atmospheric sources, which we don't have. It seems the isotope approach of Conway et al did not estimate combustion types separately. We do agree the BB contribution is probably important in big fire years in Indonesia, which is also known as the "Maritime Continent." We agree our data could inform some of the existing models developed by others.

We added new text on page 18 at line 10 and switched the order of the two sentences following the new text to improve the flow. In the final sentence about other constituents we note their importance was already discussed in Jayarathne et al. (2018).

[revised manuscript text omitted]

2. P20, L36-37: The work of Graham et al. is currently not available yet. Please provide further information on these fires, measurements, and the methodology used. Are these findings from the same, similar, or different locations/fires? Since you refer multiple times to this work in

preparation, it would be helpful if the work of Graham et al. would be available before this work is published (at least in some sort of discussion like ACPD).

**Authors response:** The paper is now published and the citations and references have been updated. In summary Graham et al. (2022) describes detailed mapping of depth of burn, rate of spread, water table, precipitation, fuel moisture, etc. obtained on 6 fires in Central Kalimantan during August – September of 2015. The fires were near the fires sampled by Stockwell et al. (2016) in Oct-Nov 2015 in Central Kalimantan and similar in general nature (e.g., on disturbed sites, near canals, etc.), but were different fires. The Graham et al study was also part of our overall NASA-supported peat fire study.

Graham, L. L. B., Applegate, G. B., Thomas, A., Ryan, K. C., Saharjo, B. H., and Cochrane, M. A.: A field study of tropical peat fire behaviour and associated carbon emissions, Fire, 5, 62, https://doi.org/10.3390/fire5030062, 2022.

3. In your study, you exclude extratropical peat fire emission data. Can you draw any conclusion from your extensive dataset to also constrain extratropical peat fire emissions or provide suggestions (e.g., for other measurement campaigns) to do so?

**Authors response:** Good point. Unfortunately, the best we can do at this time is append to the sentence on page 12 line 16 "nor do we know of extratropical field-based emissions measurements that could help identify the best lab data for this purpose."